# Cytoreductive Surgery and Hyperthermic Intraperitoneal Chemotherapy for Peritoneal Metastases from Colorectal Cancer—An Overview of Current Status and Future Perspectives

**DOI:** 10.3390/cancers16020284

**Published:** 2024-01-09

**Authors:** Wilhelm Graf, Lana Ghanipour, Helgi Birgisson, Peter H. Cashin

**Affiliations:** Uppsala Sweden and Department of Surgery, Institution of Surgical Sciences, Uppsala University, Akademiska Sjukhuset, SE-751 85 Uppsala, Sweden; lana.ghanipour@uu.se (L.G.); helgi.birgisson@uu.se (H.B.); peter.cashin@uu.se (P.H.C.)

**Keywords:** colorectal cancer, peritoneal metastases, locoregional therapy

## Abstract

**Simple Summary:**

The concept of cytoreductive surgery and hyperthermic intraperitoneal chemotherapy perfusion for the treatment of colorectal cancer peritoneal metastases has been debated based on the results of recent controlled trials. In this review, we describe the development of this “package” treatment and discuss various aspects of the selection and indications, as well as future fields of research.

**Abstract:**

Peritoneal metastases (PM) are observed in approximately 8% of patients diagnosed with colorectal cancer, either synchronously or metachronously during follow-up. PM often manifests as the sole site of metastasis. PM is associated with a poor prognosis and typically shows resistance to systemic chemotherapy. Consequently, there has been a search for alternative treatment strategies. This review focuses on the global evolution of the combined approach involving cytoreductive surgery (CRS) and hyperthermic intraperitoneal chemotherapy (HIPEC) for the management of PM. It encompasses accepted clinical guidelines, principles for patient selection, surgical and physiological considerations, biomarkers, pharmacological protocols, and treatment outcomes. Additionally, it integrates the relevant literature and findings from previous studies. The role of CRS and HIPEC, in conjunction with other therapies such as neoadjuvant and adjuvant chemotherapy, is discussed, along with the management of patients presenting with oligometastatic disease. Furthermore, potential avenues for future development in this field are explored.

## 1. Introduction

Colorectal cancer is one of the most common types of cancer, with a global incidence of 1.9 million cases per year and a worldwide death rate of 935,000 [1]. About 8% of individuals develop peritoneal metastases (PM), occurring either at presentation or during follow-up [2,3]. Peritoneal metastases present unique challenges as this metastatic site was historically associated with short survival [4], severe symptoms [5], limited extension of survival [6], and a response rate not exceeding 30% after systemic chemotherapy [7]. These factors led to a search for more effective treatments. The term cytoreductive surgery (CRS) was initially introduced in the treatment of testicular [1] and ovarian [2] tumors, based on the assumption that reducing the tumor volume enhances the effectiveness of further treatment [3,4,5]. Later, this concept was applied to the treatment of low-grade mucinous tumors originating from the colon or appendix [6]. In a specific study, complete tumor removal was followed by the intraperitoneal infusion of chemotherapy using 5-fluorouracil and mitomycin C. Five out of seven patients experienced remission following this treatment, motivating the team to further explore the combination of CRS and locoregional chemotherapy. Meanwhile, other tumor types were treated with heated chemotherapy, leveraging a pharmacokinetic advantage and the selective sensitivity of tumor cells to thermal damage, while safeguarding normal tissue through an intact cooling blood flow [7,8]. Furthermore, in vitro studies suggested the selectively increased antitumor action of chemotherapeutic compounds using hyperthermia [9]. The focus of this review is to discuss the concept of CRS and HIPEC in PM originating from large bowel cancer and to present current knowledge regarding patient selection, HIPEC regimens, predictive factors, and insights gleaned from published trials, including anticipated outcomes.

## 2. Patient Selection and Work-Up

Peritoneal metastases (PM) from colorectal cancer (CRC) occur in 8–10% of cases as either metachronous or synchronous lesions [2]. Cytoreductive surgery with hyperthermic intraperitoneal chemotherapy (CRS-HIPEC) presents a potential cure in specific patients, boasting a 5-year survival rate of 40–50% [10,11]. However, due to the elevated risk of postoperative morbidity associated with the procedure, only patients with a good performance status, age below 80 years, limited liver metastases, and favorable molecular biological characteristics may be considered for CRS-HIPEC. The extent of the disease is assessed using the peritoneal cancer index (PCI), typically evaluated intraoperatively during open exploration. The PCI score stands as one of the most acknowledged and independent prognostic factors for PM from CRC. A PCI score of 20 or less correlates with improved survival outcomes [12]. Selected patients with colorectal PM can undergo a potentially curative procedure, with survival heavily reliant on the PCI score and completeness of cytoreduction score (CCS) [13,14]. Therefore, a comprehensive diagnostic work-up plays a fundamental role in detecting PM, determining its extent, assessing metastases to other solid organs (such as the liver, lungs, extraregional lymph nodes, pleura, and bones), staging the disease, and selecting appropriate treatment strategies to enhance prognosis. Precise patient selection remains crucial in achieving long-term survival outcomes with CRS-HIPEC. Computed tomography (CT) serves as the primary imaging modality in the standard pre-operative assessment for patients considered for CRS-HIPEC [15]. CT’s sensitivity in diagnosing peritoneal metastases (PM) ranges from 60% to 94%. Crucial factors influencing this sensitivity include the lesion size [16], the specific abdominal regions examined, and the radiologist’s interpretation [15]. The detection sensitivity on CT decreases from 94% for lesions larger than 5 cm to 11% for lesions smaller than 0.5 cm [16]. Notably, the epigastrium and pelvis exhibit the highest sensitivity for the detection of PM compared to other abdominal regions [17]. CT tends to underestimate the surgical peritoneal cancer index (PCI) by 12–33% [17]. Its accuracy in assessing the small bowel serosal surfaces and the mesentery is limited, potentially leading to underdiagnosis and unsuccessful CRS.

In clinical practice, magnetic resonance imaging (MRI) serves as a secondary modality, often employed for the investigation of liver metastases or in cases involving locally advanced tumors or solid organ metastases to determine the feasibility of radical resection. PET-CT is typically used in cases concerning suspicious lymph nodes or extra-abdominal disease. MRI outperforms CT and PET-CT in detecting small tumor lesions and liver metastases [18,19]. MRI demonstrates sensitivity of approximately 80% and specificity of 85% in detecting peritoneal lesions, while also offering an approximate prediction of PCI before surgery [20]. Dohan et al. investigated the combination of CT and MRI in preoperatively estimating PCI, finding that CT along with MRI provided greater accuracy in predicting surgical PCI compared to CT alone [21]. MRI’s added value revealed increased sensitivity in detecting PM in regions such as the central quadrant, pelvis, and upper left quadrant [21]. Recent advancements, combining contrast-enhanced MRI with diffusion-weighted imaging, have resulted in a more precise description of the disease extent [22]. Preoperative staging laparoscopy is recommended as a valuable modality in patient selection for CRS-HIPEC due to the limited visualization of peritoneal metastases (PM) with noninvasive methods [23]. Staging laparoscopy offers the advantage of providing a comprehensive view of all regions, allowing the assessment of the small bowel serosal surface and mesentery. It aids in excluding patients with extensive peritoneal disease from exploratory laparotomy or an open/closed procedure, particularly when palliative chemotherapy is a better alternative. Iversen et al. demonstrated that 40% of patients were spared from exploratory laparotomy due to extensive peritoneal disease identified during staging laparoscopy, although 17% were deemed inoperable at the time of laparotomy [23]. However, adhesions or tumor masses may limit the thorough evaluation of all intraabdominal regions during laparoscopy. Despite occasional limitations in visualization and the associated risks of port metastases and minor surgical complications linked to the invasive nature of diagnostic laparoscopy, it is considered a safe and reliable modality in the preoperative management of patients with PM from CRC [23]. The inclusion of diagnostic laparoscopy in preoperative assessments may be beneficial, especially when CT shows a borderline PCI score [24]. Apart from tumor extension, patient related factors like age, comorbidity, and fragility should be taken into account in the selection process, but these aspects play a similar role as in all major abdominal procedures. In addition, the tumor biomarkers presented below could have important impacts on the selection of patients for CRS and HIPEC.

## 3. Prognostic and Predictive Factors

Undoubtedly, a noninvasive preoperative modality with high sensitivity, specificity, and accuracy in detecting PM is crucial for the selection of eligible patients for CRS-HIPEC. An ongoing multicenter randomized controlled trial in Holland aims to investigate whether MRI can replace staging laparoscopy as the preoperative modality for patients with PM from CRC eligible for CRS-HIPEC [25]. Despite careful patient selection, the majority of patients with CRC-PM will eventually develop recurrent disease. Known prognostic factors other than the PCI score and CC score with a negative impact on overall survival are locoregional lymph node metastases, a low differentiation grade, and the presence of a signet ring cell histology [26]. The peritoneal surface disease severity score (PSDSS) and the colorectal peritoneal metastases prognostic surgical score (COMPASS) are validated prognostic nomograms used as a clinical scores for the prediction of patient survival after CRS-HIPEC [26,27]. These nomograms are based on the age, PCI score, loco-regional lymph node status, and presence of signet ring cells. However, these nomograms do not consider the information from molecular markers. Nonetheless, other than clinical factors, there is also a need for a better understanding of molecular factors in relation to tumor biology in the selection process for CRS-HIPEC. In recent years, the classification of molecular markers has gained increased awareness in order to create a more personalized therapy in mCRC. Well-known predictive markers are the mutation status of the oncogenes KRAS, NRAS, and HRAS. Mutations in these oncogenes result in the constitutive activation of the Ras-Raf-MAPK pathway with the dysregulation of the cellular proliferation of epidermal growth factor receptor (EGFR). KRAS mutations are found in 40–46% and BRAF mutations in 5–11% of all mCRC cases [28,29]. It is well known that mutations in these genes have no beneficial response rate to anti-EGFR monoclonal antibody therapy [28]. Mutations in KRAS and BRAF are shown to have a negative impact on survival after CRS-HIPEC, independently of anti-EGFR antibody therapy [30]. Mutations in the mismatch repair (MMR) system, known as defective mismatch repair (dMMR), occur in 10–20% of all sporadic CRC cases [31]. An understanding of the mismatch repair status has provided deeper insights into the heterogeneity of CRC biology, making it a valuable prognostic and predictive marker. dMMR tumors have demonstrated a poorer response to 5-fluorouracil-based chemotherapy [32,33]. Compared to proficient mismatch repair (pMMR) tumors, dMMR tumors exhibit a distinct recurrence pattern with a higher local recurrence rate (30% vs. 12%) and a tendency toward peritoneal metastasis (40% vs. 12%), leading to an overall worse prognosis upon recurrence [34]. These tumors possess reduced metastatic potential, observed in 3–5% of cases [31], and are often linked with BRAF mutations (35%) and a poorer prognosis [35]. In a multicenter study, Tonello et al. evaluated the prognostic role of MMR and the RAS/RAF mutation status in CRC-PM patients undergoing CRS-HIPEC. They found that patients with dMMR had better overall survival compared to pMMR (5-year OS, 58% vs. 37%), and the most favorable prognosis was observed in those with dMMR and KRAS/BRAF wildtype [32]. Additionally, the presence of KRAS and BRAF mutations negatively impacted not only overall survival but also disease-free survival [36]. The emerging predictive role of dMMR in mCRC has garnered attention since the introduction of immune checkpoint inhibitors. In a randomized phase 3 trial, the programmed death-1 (PD-1) inhibitor pembrolizumab demonstrated a superior impact on progression-free survival in patients with dMMR mCRC compared to chemotherapy. This suggests that pembrolizumab should be considered as an initial therapy option in dMMR mCRC [37]. However, long-term outcomes with immune checkpoint inhibitors in mCRC remain inadequately studied, necessitating further research. Understanding the molecular changes leading to recurrent and metastatic disease remains a significant challenge. A deeper comprehension of genetic alterations to identify potentially prognostic and predictive biomarkers for personalized cancer treatment is crucial for optimal patient selection. Further research in this realm is highly warranted to advance personalized cancer therapies.

## 4. The Rationale of HIPEC Treatment—Pharmacokinetics and Hyperthermia

The roots of intraperitoneal (IP) therapy trace back to 1744, when English surgeon Christopher Warrick utilized “Bristol water” and Bordeaux wine in the peritoneal cavity to address intractable ascites [38]. However, it was not until the late 1970s that IP therapy gained substantial traction, primarily in ovarian cancer cases [39]. During this period, it became evident that IP administration offered specific pharmacokinetic advantages. It allowed the delivery of chemotherapy in higher doses, yielding a more potent locoregional effect within the abdominal cavity. Early on, the challenge was recognized—the drug’s penetration was rather limited, usually around 1–2 mm depending on the specific medication [40,41]. For IP therapy to exhibit promise, it either needed to be combined with the removal of larger tumor nodules during cytoreductive surgery or administered as repeated intermittent treatments over an extended duration to affect macroscopic tumor nodules. Cytoreductive surgery, with HIPEC treatment as an adjunct, emerged as a solution for the former situation. Alternatively, Japan primarily utilized repeat IP treatments preoperatively to treat macroscopic peritoneal metastases from gastric cancer, awaiting the disappearance of peritoneal nodules before performing a final gastrectomy [42]. Similar repeat treatments have been employed postoperatively as an adjuvant therapy after ovarian or colorectal cancer surgeries with peritoneal metastases [43,44]. Presently, HIPEC is the most globally utilized mode of treatment. However, postoperative IP use has largely ceased due to issues arising from postoperative adhesions, hindering planned treatments for many patients [43,44]. IP catheter complications also pose challenges to this treatment modality. Preoperative or palliative IP use remains an area of significant investigation, particularly with the emergence of pressurized intraperitoneal aerosolized chemotherapy (PIPAC) treatment [45]. This review will not delve into this emerging research field, but numerous ongoing trials are exploring this new IP modality. Further results are expected to emerge in the upcoming years.

### 4.1. Pharmakokinetics

As previously highlighted, the pharmacokinetic advantage of achieving higher intraperitoneal concentrations has been well demonstrated in various studies. One method to assess this advantage is by calculating the intraperitoneal to systemic exposure ratio, determined by dividing the area under the concentration curve (AUC) intraperitoneally by the AUC in plasma. Depending on the drug and perfusion time, this AUC peritoneum/AUC plasma ratio can range widely, from 8 to 1000 [46]. However, it is crucial to note that this ratio alone does not fully represent the chemotherapy uptake in peritoneal nodules. Several drug-related factors significantly contribute to determining how much of the compound is transported into the tumor nodule, aside from the concentration gradient created by this ratio (see Table 1) [47].

The passage of a drug into the tumor nodule occurs through two mechanisms—convection and diffusion. Convection relies on the pressure disparity between the fluid-filled cavity and the stromal tissue pressure. The drug’s velocity is invariably slower than the carrier fluid in which it is dissolved, forming the basis of the retardation coefficient. Notably, tumor stromal tissue often experiences heightened pressure compared to normal tissue, partly due to increased interstitial fluid pressure and various other factors outlined in Table 1. Hyperthermia, such as that used in HIPEC, and elevating intraabdominal pressure, as in PIPAC, are approaches employed to influence this pressure differential. On the other hand, diffusion relies on the concentration gradient. Theoretically, drugs with a high AUC peritoneum/AUC plasma ratio can enhance diffusion into the tumor nodule. However, several crucial stromal properties can impact diffusion. These include the viscoelasticity or stiffness of the tumor nodule, as well as the density and geometric arrangement of fibers. For more comprehensive details of these factors, refer to Table 1 in a recent review [47].

### 4.2. Hyperthermia

Certainly, hyperthermia possesses dual effects on malignant cells. While it is recognized for its potential lethality to cancer cells [48], hyperthermia can act as a double-edged sword. It triggers the induction of heat shock proteins that, under certain circumstances, may exert anti-apoptotic and proliferative effects on tumor cells [49,50]. The clinical significance of these effects remains uncertain.

Nevertheless, extensive research has investigated the synergy between hyperthermia and the enhanced uptake of chemotherapeutic drugs [9,51,52,53,54]. Platinum compounds have consistently demonstrated a synergy with hyperthermia, whereas mitomycin C has shown conflicting results. Conversely, certain compounds like taxanes have shown no enhancement with hyperthermia. In a rat study, hyperthermia alone and mitomycin C alone impeded peritoneal metastatic growth, but their combination had a notably synergistic effect, surpassing the efficacy of either treatment alone [55]. Regarding clinical trials, there is a single randomized clinical trial evaluating the use of hyperthermia in gastric cancer [56]. In this trial, patients with gastric cancer and peritoneal metastases underwent gastrectomy alone (surgery alone arm), gastrectomy with normothermic intraperitoneal chemotherapy (NIPEC) at 37 degrees, or gastrectomy with hyperthermic intraperitoneal chemotherapy (HIPEC) at 41–42 degrees. Certainly, in the mentioned study, the multivariable model demonstrated a hazard ratio of 1.77 (95% confidence interval 0.91–3.42, *p* = 0.092) for the use of hyperthermia. Although this result did not reach full statistical significance, it indicated an intriguing trend. This underscores the need for further research to delve deeper into the clinical utility of hyperthermia. Currently, HIPEC stands as the most prevalent intraperitoneal (IP) treatment modality worldwide [57]. Despite its widespread use, numerous unanswered questions persist regarding its pharmacological and hyperthermic rationale. There is a compelling need for additional research to understand better the effects of HIPEC on the tumor microenvironment. Furthermore, there is a crucial necessity for more randomized clinical trials to evaluate the various components and intricacies of HIPEC treatment. Such research endeavors would significantly contribute to elucidating its efficacy and guiding its optimal utilization in clinical settings.

## 5. Results from Published Trials

Certainly, the treatment landscape for isolated colorectal peritoneal metastasis (CRPM) has evolved significantly. It is widely acknowledged that the survival of patients solely treated with systemic chemotherapy for isolated colorectal PM is limited, typically resulting in median overall survival (OS) of around 16 months [4]. Moreover, the prognosis is notably worse for patients with PM along with other metastatic sites compared to those with isolated PM. Cytoreductive surgery (CRS) has emerged as an increasingly utilized approach for PM, offering the possibility of extended survival and even a potential cure. However, randomized studies examining the effect of CRS and HIPEC for colorectal peritoneal metastasis (CRPM) remain scarce, with retrospective studies dominating the available literature (Table 2). 

Two published randomized trials shed light on the benefits of CRS combined with locoregional chemotherapy compared to systemic chemotherapy without CRS [44,58]. In a Dutch study conducted between 1998 and 2001, 105 patients were randomized. One group received standard treatment involving systemic chemotherapy with 5-FU and leucovorin (*n* = 51), while the other group underwent CRS and HIPEC involving MMC (*n* = 54). In the standard treatment arm, 43 patients completed the planned 6 months of treatment, while, in the experimental arm, only 41% achieved complete cytoreduction; however, all received HIPEC. Over an eight-year follow-up period, disease-specific survival was notably different between the two groups, with the standard arm showing survival of 12.6 months and the experimental arm demonstrating significantly longer survival of 22.2 months (*p* = 0.028). Notably, patients who underwent complete cytoreduction had significantly improved survival compared to those who had incomplete cytoreduction [58]. The Swedish study randomized patients with colorectal peritoneal metastases (PM) to either systemic chemotherapy with FOLFOX for 6 months or to cytoreductive surgery followed by intraperitoneal chemotherapy of 5-FU and leucovorin given through an abdominal port catheter within 3 h after surgery. The planned enrollment was 100 patients, but, due to slow accrual, the study was halted prematurely after 7 years (2004–2011), including 24 eligible patients in each arm. Within the surgery arm, 14 patients (58%) achieved complete cytoreduction. The 5-year overall survival (OS) was notably higher in the surgery arm at 33% (*n* = 8), compared to 4% (*n* = 1) in the systemic chemotherapy arm. As observed in prior studies, survival was notably improved among those who underwent complete cytoreduction compared to those with incomplete cytoreduction [44]. In addition, two well-designed comparative but non randomized studies suggested a definite survival advantage after CRS and HIPEC compared with systemic chemotherapy alone (Table 2).

These studies have significantly contributed to our understanding of the optimal therapy for patients with PM, indicating that chemotherapy alone provides a limited likelihood of extended survival. Notably, cytoreductive surgery demonstrates its benefit particularly in patients achieving complete cytoreduction. Presently, cytoreductive surgery is typically pursued only when complete cytoreduction is deemed feasible, as multiple studies have highlighted its significance in minimizing the recurrence risk and prolonging survival. The completeness of cytoreduction score (CCS) was developed to categorize the extent of CRS, with CCS = 0 indicating complete cytoreduction [13,14,64]. Moreover, it was demonstrated that surgery and intraperitoneal chemotherapy were not associated with more severe treatment-related complications compared to systemic chemotherapy [44].

The recent French multicenter randomized trial (PRODIGE 7) examining the impact of HIPEC in addition to cytoreductive surgery (CRS) raised questions about the actual effect of HIPEC as compared to the significance of CRS itself [60]. During the period from 2008 to 2014, patients with colorectal peritoneal metastases (CRPM) were randomized into two groups: CRS alone (*n* = 132) or CRS followed by HIPEC using oxaliplatin intraperitoneally and 5-FU and leucovorin intravenously (*n* = 133). The majority of patients received neoadjuvant chemotherapy with a median of six cycles. Randomization was done perioperatively, excluding patients with PCI > 25, irresectable disease, liver metastasis, and no macroscopic peritoneal disease. Both arms achieved a 90% complete cytoreduction rate. The 5-year follow-up showed a median OS of 41.2 months in the CRS group and 41.7 months in the CRS and HIPEC group. Notably, 16 patients in the CRS arm were later treated with CRS + HIPEC upon developing isolated peritoneal recurrences. Additionally, the subgroup analysis revealed that patients with a PCI of 11–15 had significantly better OS if treated with CRS + HIPEC rather than CRS alone [60].

Although the peritoneal cancer index (PCI) is a widely used tool to estimate the peritoneal tumor burden, it has limitations, including low interobserver agreement and the potential overestimation of malignant peritoneal disease due to difficulties in differentiating benign fibrotic lesions from true malignant colorectal peritoneal metastases [65,66,67]. Therefore, caution should be exercised when considering using PCI alone, particularly focusing on PCI 11–15, as an indicator for HIPEC in addition to CRS.

Variations in the application of HIPEC in addition to CRS for PM exist internationally, as highlighted in a recent paper by the Peritoneal Surface Oncology Group International (PSOGI) [68]. Methodological differences, drug regimens, dosages, and other variations among institutions make direct comparisons between studies challenging. Presently, in the Nordic countries, HIPEC is considered an integral part of the treatment protocol along with CRS for PM.

The idea of using HIPEC as a prophylactic treatment to reduce the risk of PM in high-risk colorectal cancer patients has been debated. Certain factors like T4 disease, N2 involvement, right-sided tumors, vascular invasion, mucinous tumors, and emergency surgery result in a higher risk of developing metachronous PM [2,69]. However, implementing prophylactic HIPEC in such cases might result in the overtreatment of those who do not develop PM, as noted by Arrizabalaga et al. [70]. Despite this, a recent randomized trial indicated a reduced risk of local recurrence after prophylactic HIPEC in T4 tumors [63]. Hence, the utility of prophylactic HIPEC in high-risk colorectal cancer patients remains an area for discussion and further research.

Certainly, studies evaluating preemptive treatments, particularly second look surgeries aiming for cytoreductive surgery (CRS) and HIPEC in patients at high risk of developing metachronous colorectal peritoneal metastases (CRPM), have been conducted. The COLOPEC trial randomized patients with T4N0-M0 stage or perforated colon cancer into two groups: one receiving HIPEC followed by adjuvant chemotherapy (experimental group) and the other receiving adjuvant chemotherapy alone after resection of the colonic cancer (control group) [62]. The primary endpoint was PM-free survival at 18 months, assessed using diagnostic laparoscopy for patients free of disease recurrence. No significant difference was observed in peritoneal-free survival at 18 months between the two groups (80.9% [95% CI 73.3–88.5] for the experimental group vs. 76.2% [68.0–84.4] for the control group; *p* = 0.28).

Similarly, the PROPHYLOCHIP study, a French multicenter trial, randomized patients with primary colorectal cancer and synchronous localized CRPM, resected ovarian metastasis, or perforated tumors to either second look surgery after 6 months with HIPEC (experimental group) or surveillance only (control group) [61]. The primary outcome was 3-year disease-free survival, which showed rates of 53% (95% CI 41–64) in the control arm versus 44% (33–56) in the experimental group (HR 0.97; 95% CI 0.61–1.56).

Based on the outcomes of these trials, the standard of care for curatively treated patients with colorectal cancer at high risk of developing PM currently involves surveillance utilizing radiology and tumor markers. These studies did not demonstrate a significant benefit from preemptive treatments such as second look surgeries with HIPEC in improving PM-free survival or disease-free survival at 18 months or 3 years, respectively.

The existing data, while limited, emphasize the significance of complete cytoreduction during CRS as a crucial factor in determining patient outcomes. However, further studies exploring various aspects of HIPEC, such as the optimal types, dosages, and combinations of chemotherapeutic agents utilized, are essential. More comprehensive research in this domain is necessary to refine and establish HIPEC protocols that can potentially enhance the treatment efficacy and improve patient outcomes in CRPM cases.

## 6. Patterns of Recurrence

The risk for recurrence after CRS and HIPEC for CRPM is high, with 5-year progression-free survival expected to be less than 20% and median progression-free survival of 15 months in 660 patients treated in Netherlands [71]. Breuer et al. revealed that in 505 patients treated with CRS and HIPEC for CRPM and having a median PCI of 6, 71.5% developed recurrences, 28.3% developed isolated hematogenous metastases, 24.6% had isolated PM, and 13.9% had mixed peritoneal and hematogenous metastases [72]. Those with isolated or mixed peritoneal metastases had a shorter time to recurrence than those with isolated hematogenous metastases, with hepatic and pulmonary metastases the most common hematogenous metastatic sites. Braams et al. revealed that out of 132 patients having recurrent disease after CRS and HIPEC, 32 underwent resection of the metastases, of which 17 were locoregional and 14 distal; it was more likely that the metastasis was resectable if the interval between the index CRS and HIPEC and recurrence was prolonged [73]. The recurrence risk is dependent on several factors. Previously, we have mentioned the completeness of cytoreduction, with CCS = 0 being the only group that can expect a cure or long-term disease free survival. The PCI is also of importance as a high PCI score is associated with a greater risk of recurrence. In the PRODIGE 7, study those with PCI < 11 had 23% DFS after 3 years, compared with 4% for those with PCI 11–15 and 3% if PCI > 15. Other factors, such as advanced *n* stage [72], signet ring cell differentiation [26], BRAF mutation [29], and gains of chromosome 1p and 15q [74], also have negative effects on the prognosis. A recent study suggests that BRAF mutations also increase the risk of metachronous peritoneal metastasis in colon cancer patients and thereby also imply an increased risk of recurrence after CRS and HIPEC [75].

## 7. Conclusions

The efficacy of hyperthermic intraperitoneal chemotherapy (HIPEC) in conjunction with cytoreductive surgery (CRS) for colorectal cancer-related peritoneal metastases has been debated, despite being a standard treatment option. While some studies have raised doubts about the significant contribution of HIPEC to treatment outcomes, CRS plus HIPEC remains a primary therapeutic approach for peritoneal spread from colorectal cancer. Recent research and emerging studies have emphasized the need for ongoing efforts to optimize the patient selection criteria and refine the administration of chemotherapy in HIPEC. This includes exploring modifications in the selection process or adjusting the dosage of chemotherapy agents used during HIPEC. The goal is to enhance the efficacy of the treatment by improving the patient selection parameters and refining the delivery of chemotherapy within the peritoneal cavity to achieve better tumor eradication and control.

## Figures and Tables

**Table 1 cancers-16-00284-t001:** Drug- and tumor-related factors influencing tumor uptake.

IP Drug Properties	Tumor Microenvironment
Concentration	Interstitial fluid pressure
Molecular weight	Solid pressures
Hydrodynamic diameter	Hydraulic conductivity
Configuration	Viscoelasticity, stiffness
Water solubility	Retardation coefficient
Protein binding	Cellular composition
Charge, ionization	Stromal and vascular density
	Geometrical arrangement

**Table 2 cancers-16-00284-t002:** Numbers and survival differences in randomized and comparative studies examining the role of CRS and HIPEC for treatment or prevention of peritoneal metastases.

Treatment of PM	Chemotherapy Alone	CRS + IPC/HIPEC	*p*-Value
Randomized			
Verwaal et al. [58]	51	54	0.028
Cashin et al. [44]	24	24	0.04
Non-randomized			
Franko et al. [59]	38	67	<0.001
Elias et al. [10]	48	48	<0.05
Treatment of PM	CRS Alone	CRS + HIPEC	*p*-value
PRODIGE 7 [60]	132	133	0.99
Prophylactic Treatment	Control Group	Prophylactic HIPEC	*p*-value
Prophylochip [61]	75	75	0.82
COLOPEC [62]	102	102	0.28
HIPECT4 [63]	95	89	0.68

## Data Availability

Not applicable.

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
