# Peer review of "Cytoreductive Surgery and Hyperthermic Intraperitoneal Chemotherapy for Peritoneal Metastases from Colorectal Cancer—An Overview of Current Status and Future Perspectives"

_cancers, 2024, doi:10.3390/cancers16020284_

Round 1

Reviewer 1 Report

Comments and Suggestions for Authors

 Cytoreductive Surgery and Hyperthermic Intraperitoneal Chemotherapy for Peritoneal Metastases from Colorectal Cancer—An Overview of Current Status and Future Perspectives, makes a case for hyperthermic intraperitoneal chemotherapy following cytoreductive surgery.  The paper does present a strong argument for CRS and leaves the use of HIPEC somewhat speculative.  In fact, patient pre-therapy screening seems to be a major issue here, limiting the numbers of patients might benefit and would actually be considered for such treatment.

In the Conclusion section, please remove the statement the “authors still believe”, and replace with a statement to the effect that you are interested in pursuing this line of treatment and the reasons for your interest—maybe keeping options open for patients in specific cases, as mentioned on lines 290-292, or other reasons.

Stylistically, lines 151-158 should be moved up to line 135.

There are a few spelling, and grammar errors, throughout the paper.  The header on line 52 is misplaced.

On line 150, what is meant by “dMMR/wildtype?- hets?

Line 159 far(e) far?

Line 259 randomize(d)

Line 264 was should be were

Line 277 chemo(t)herapy

Line 288 Five y(ear)

Linea 297-298 should be moved up to line 292.

Line 368 “A conclusion and”  remove

Comments on the Quality of English Language

needs work

Author Response

We thank the Reviewer for the efforts to improve our manuscript. Please find detailed responses below and the corresponding revisions/corrections highlighted/in track changes in the revised version.

Cytoreductive Surgery and Hyperthermic Intraperitoneal Chemotherapy for Peritoneal Metastases from Colorectal Cancer—An Overview of Current Status and Future Perspectives, makes a case for hyperthermic intraperitoneal chemotherapy following cytoreductive surgery.  The paper does present a strong argument for CRS and leaves the use of HIPEC somewhat speculative.  In fact, patient pre-therapy screening seems to be a major issue here, limiting the numbers of patients might benefit and would actually be considered for such treatment.

Thanks for this comment. We have tried to make it even more clear in the revised version that the patient selection process is of utmost importance for obtaining successful outcome after CRS and HIPEC.

In the Conclusion section, please remove the statement the “authors still believe”, and replace with a statement to the effect that you are interested in pursuing this line of treatment and the reasons for your interest—maybe keeping options open for patients in specific cases, as mentioned on lines 290-292, or other reasons.

This statement has been removed from the conclusion.

Stylistically, lines 151-158 should be moved up to line 135.

In the revised version the predictive importance of MMR and RAS status are presented in an integrated fashion.

There are a few spelling, and grammar errors, throughout the paper.  The header on line 52 is misplaced.

This error has been corrected. Spelling and grammar has been carefully checked and the revised version has been subjected to a linguistic revision.

On line 150, what is meant by “dMMR/wildtype?- hets?

We apologize for this mistake. dMMR/KRAS, BRAF wildtype is the correct wording and has been changed in the revised version.

Line 159 far(e) far?

Corrected in revised version.

Line 259 randomize(d)

Corrected in revised version.

Line 264 was should be were

Corrected in revised version.

Line 277 chemo(t)herapy

Corrected in revised version.

Line 288 Five y(ear)

Corrected in revised version.

Linea 297-298 should be moved up to line 292.

Maybe not applicable after linguistic revision.

Line 368 “A conclusion and”  remove

Corrected in revised version.

Comments on the Quality of English Language

needs work

Spelling and grammar has been carefully checked and the revised version has been subjected to a linguistic revision.

Reviewer 2 Report

Comments and Suggestions for Authors

The paper addressed highly important issue, i.e., it discusses the role of HIPEC in colorectal cancer management. To my opinion, the message of the paper needs to be articulated in a more straightforward way. The biological rationale, the existing controversies and the preclinical data have to be discussed in the beginning of the paper. The clinical data require proper interpretation: which categories of patients may benefit from HIPEC, and in for which patients this intervention is not feasible and/or contraindicated? It is advisable to present main clinical trials and their conclusions in a table format. Data on molecular markers are confusing and misleading: they may be mentioned only with the reference to HIPEC trials, with a numbers and statistical calculations provided.  The Conclusion section is vague: it does not provide either advice for the doctors or avenue for future research. The Abstract does not deliver the message of the paper: please provide a concise summary of practical recommendations and list the existing controversies.  There are some typing errors, so careful examination of the text is needed.

Comments on the Quality of English Language

 Minor editing is required. There are some typing errors, so careful examination of the text is needed.

Author Response

Comments and Suggestions for Authors

We thank the Reviewer for the efforts to improve our manuscript. Please find detailed responses below and the corresponding revisions/corrections highlighted yellow in the revised version.

The paper addressed highly important issue, i.e., it discusses the role of HIPEC in colorectal cancer management. To my opinion, the message of the paper needs to be articulated in a more straightforward way. The biological rationale, the existing controversies and the preclinical data have to be discussed in the beginning of the paper. The clinical data require proper interpretation: which categories of patients may benefit from HIPEC, and in for which patients this intervention is not feasible and/or contraindicated? It is advisable to present main clinical trials and their conclusions in a table format. Data on molecular markers are confusing and misleading: they may be mentioned only with the reference to HIPEC trials, with a numbers and statistical calculations provided.  The Conclusion section is vague: it does not provide either advice for the doctors or avenue for future research. The Abstract does not deliver the message of the paper: please provide a concise summary of practical recommendations and list the existing controversies.  There are some typing errors, so careful examination of the text is needed.

Thank You for this constructive criticism. The manuscript has been restructured to comply with the comments including the abstract, introduction and conclusion. The section on molecular markers has been rewritten to be more precise which studies relate to CRS and HIPEC patients. In addition, a new Table 2 summarizing results of main clinical trials has been added.

Comments on the Quality of English Language

 Minor editing is required. There are some typing errors, so careful examination of the text is needed.

Spelling and grammar has been carefully checked and the revised version has been subjected to a linguistic revision.

Reviewer 3 Report

Comments and Suggestions for Authors

Dear authors.

It has been a pleasure to review the present manuscript.

I think it is pretty well summarized and stratified, although I have a couple of suggestions to improve the manuscript:

- The possibility of adding a table with the summarized outcomes of the most representative studies in the field (PROPHYLOCHIP, COLOPEC, HIPECT4, PRODIGE 7 and every studies considered relevant).

 I would suggest to do a new Pubmed search and incorporate the most updated references. There are some important papers missing from the last year regarding diagnosis and risk stratification for metachronous peritoneal metastasis in colon cancer patients.

Best regards.

Comments on the Quality of English Language

Minor spelling mistakes. 

Please check carefully.

Author Response

I think it is pretty well summarized and stratified, although I have a couple of suggestions to improve the manuscript:

We thank the Reviewer for the efforts to improve our manuscript. Please find detailed responses below and the corresponding revisions/corrections highlighted yellow in the revised version.

- The possibility of adding a table with the summarized outcomes of the most representative studies in the field (PROPHYLOCHIP, COLOPEC, HIPECT4, PRODIGE 7 and every studies considered relevant).

Thank You for this suggestion, a new Table 2 summarizing results of main clinical trials has been added.

-  I would suggest to do a new Pubmed search and incorporate the most updated references. There are some important papers missing from the last year regarding diagnosis and risk stratification  for metachronous peritoneal metastasis in colon cancer patients.

We have performed a new pubmed search specifically addressing your suggestion. A recent publication was added to the reference list.

Comments on the Quality of English Language

Minor spelling mistakes. 

Please check carefully.

Spelling and grammar has been carefully checked and the revised version has been subjected to a linguistic revision.

Round 2

Reviewer 1 Report

Comments and Suggestions for Authors

Much improved and acceptable for publication.

Reviewer 2 Report

Comments and Suggestions for Authors

The authors have improved their paper in response to the critical comments.

Comments on the Quality of English Language

Minor editing is required.